# Mildly Pasteurized Whey Protein Promotes Gut Tolerance in Immature Piglets Compared with Extensively Heated Whey Protein

**DOI:** 10.3390/nu12113391

**Published:** 2020-11-04

**Authors:** Marit Navis, Lauriane Schwebel, Susanne Soendergaard Kappel, Vanesa Muncan, Per Torp Sangild, Evan Abrahamse, Lise Aunsholt, Thomas Thymann, Ruurd M. van Elburg, Ingrid B. Renes

**Affiliations:** 1Tytgat Institute for Intestinal and Liver Research, Amsterdam Gastroenterology Endocrinology and Metabolism, Amsterdam UMC, University of Amsterdam, 1105 BK Amsterdam, The Netherlands; m.navis@amsterdamumc.nl (M.N.); v.muncan@amsterdamumc.nl (V.M.); 2Danone Nutricia Research, 3584 CT Utrecht, The Netherlands; lauriane.schwebel@external.nutricia.com (L.S.); evan.abrahamse@danone.com (E.A.); 3Department of Veterinary and Animal Sciences, Comparative Pediatrics & Nutrition, University of Copenhagen, DK-1870 Copenhagen, Denmark; susanne.soendergaard.kappel@regionh.dk (S.S.K.); pts@sund.ku.dk (P.T.S.); lise.aunsholt@regionh.dk (L.A.); thomas.thymann@sund.ku.dk (T.T.); 4Department of Neonatology, Rigshospitalet, DK-2100 Copenhagen, Denmark; 5Department of Pediatrics, Odense University Hospital, DK-5000 Odense, Denmark; 6Laboratory of Food Chemistry, Wageningen University, 6708 PD Wageningen, The Netherlands; 7Emma Children’s Hospital, Amsterdam UMC, University of Amsterdam, 1105 AZ Amsterdam, The Netherlands; rm.vanelburg@amsterdamumc.nl

**Keywords:** infant milk formula, whey proteins, thermal processing, digestion, gastro-intestinal tolerance

## Abstract

Human milk is the optimal diet for infant development, but infant milk formula (IMF) must be available as an alternative. To develop high-quality IMF, bovine milk processing is required to ensure microbial safety and to obtain a protein composition that mimics human milk. However, processing can impact the quality of milk proteins, which can influence gastro-intestinal (GI) tolerance by changing digestion, transit time and/or absorption. The aim of this study was to evaluate the impact of structural changes of proteins due to thermal processing on gastro-intestinal tolerance in the immature GI tract. Preterm and near-term piglets received enteral nutrition based on whey protein concentrate (WPC) either mildly pasteurized (MP-WPC) or extensively heated (EH-WPC). Clinical symptoms, transit time and gastric residuals were evaluated. In addition, protein coagulation and protein composition of coagulates formed during in vitro digestion were analyzed in more detail. Characterization of MP-WPC and EH-WPC revealed that mild pasteurization maintained protein nativity and reduced aggregation of β-lactoglobulin and α-lactalbumin, relative to EH-WPC. Mild pasteurization reduced the formation of coagulates during digestion, resulting in reduced gastric residual volume and increased intestinal tract content. In addition, preterm piglets receiving MP-WPC showed reduced mucosal bacterial adherence in the proximal small intestine. Finally, in vitro digestion studies revealed less protein coagulation and lower levels of β-lactoglobulin and α-lactalbumin in the coagulates of MP-WPC compared with EH-WPC. In conclusion, minimal heat treatment of WPC compared with extensive heating promoted GI tolerance in immature piglets, implying that minimal heated WPC could improve the GI tolerance of milk formulas in infants.

## 1. Introduction

Human milk supports the postnatal development and maturation of the gastro-intestinal tract and is optimal for infant nutrition [1,2,3,4]. When human milk is not available, infant milk formula (IMF) derived from bovine milk is usually provided as an alternative. To obtain IMF, processing of bovine milk is required to ensure microbial safety and to more accurately mimic the composition of human milk, as the whey/casein ratio in human milk is different from bovine milk. Caseins are partly digested in the stomach and have a high nutritive value, thereby supporting the growth of the infant. Whey proteins are less readily digested and besides delivery of essential amino acids and nitrogen, they have important bioactive functions supporting gut development, immunomodulation and anti-bacterial properties [5,6,7]. The major whey proteins are β-lactoglobulin and α-lactalbumin [8]. The whey to casein ratio in human milk is 60:40 compared with 20:80 in bovine milk [9]. In addition, human milk has less κ-casein and no α-casein compared with bovine milk [10]. By processing bovine milk, a whey and casein ratio closer to that in human milk can be obtained [11].

Besides quantity of the specific proteins, the quality of these proteins in the milk are important for short-term and long-term development. Historically, infant gastro-intestinal (GI) tolerance of IMF appeared to be lower than for human milk [12], which led to various efforts to improve the quality of IMF. Feeding intolerance is defined as increased gastric residuals, abdominal distension, regurgitation, emesis and diarrhea, together reflecting an inability to digest milk properly [13]. Feeding intolerance is frequently observed in premature infants due to immaturity of the GI tract, and can result in secondary problems like malabsorption, impaired growth and intestinal infections [13]. Milk processing can influence protein digestibility, gastric emptying and GI transit time and thereby GI tolerance. For example, proteins in bovine milk can form aggregates upon heating, which can result in lower protein bioavailability and delayed gastric emptying [14]. In addition, heating can also cause protein denaturation, leading to changes in bioactivity and/or protein digestibility [14]. Another potential heat-induced change in protein structure can be initiated by Maillard reaction glycation, when lysine is glycated with lactose, which also will impact bioavailability and bioactivity of those proteins [15].

Traditional milk processing methods used in IMF production involve rennet or acid precipitation to separate the soluble whey protein fraction from the precipitated casein. Cold membrane filtration of skim milk is a relatively novel way of processing that results in a protein base ingredient for infant formula with a casein-to-whey ratio and a casein profile that more closely resembles human milk. Specifically, it eliminates α-casein and κ-casein, which are present in bovine milk but absent or only in low abundance in human milk [16]. In addition, as this improved way of processing only applies minimal heating, the milk proteins are kept in their native form thereby limiting the impact on protein characteristics. 

Piglets are widely used as a preclinical model in nutrition research, based on the high similarities of the porcine and human GI tract considering morphology, physiology, metabolism and innate defense [17,18]. The preterm piglet model specifically shows high sensitivity to dietary interventions, induced by a combination of preterm birth, cesarean delivery and deprivation of a sow’s colostrum which contains many bioactive peptides that stimulate gut maturation and protect against gut inflammation [19,20]. The preterm piglet model has been used to study gut maturation and necrotizing enterocolitis (NEC) in relation to diet. Recently, it was shown that aspects of feeding tolerance can also be evaluated in this model, using gastric residual measurements and X-ray contrast imaging [21].

The current study investigated GI tract tolerance to mildly pasteurized whey protein concentrate (MP-WPC) compared with more extensively heated WPC (EH-WPC) in premature piglets, which are highly sensitive to dietary interventions. Secondly, a group of near-term piglets was included to explore whether the impact on GI tolerance was similar in piglets delivered at later gestational age, as gut structure and function develop rapidly in piglets in the last weeks before the term date [22]. The aim was to identify how structural modifications of proteins during thermal processing may affect nutritional and functional properties, specifically related to aspects of GI tolerance, using piglets as a model for infants. 

## 2. Materials and Methods 

### 2.1. Whey Protein Concentrate (WPC)

WPC was prepared from fresh cow’s milk by cold membrane filtration (0.08 µm pore size) and spray drying on a pilot scale, as described previously [16]. The WPC was dissolved in water and mildly pasteurized for 30 s at 73 °C, after which it was freeze-dried. A detailed overview of all processing steps included is provided in Figure 1a.

WPC solutions tested in vivo and in vitro were prepared fresh daily. MP-WPC powder was dissolved in demineralized water at 100 g/L, corresponding to 68.3 g protein per liter. pH was adjusted to 7.1 by addition of 1M NaOH, to prevent gelation during heating. Half of the MP-WPC solution was used without further processing (MP-WPC). The other part was thermally treated to denature whey protein using a shaking water bath at 80 °C for 20 min: extensively heated-WPC (EH-WPC). Temperature was monitored and reached 80 °C after 14 min, thus the effective thermal treatment at 80 °C was 6 min. Following thermal treatment, solutions were cooled to room temperature (RT) for at least 40 min before any further testing. Protein compositions as determined by SDS-page are described in Table 1.

To determine the soluble protein fraction, WPC protein solution at pH 7.1 was centrifuged at 15,000× *g* for 30 min and the supernatant was collected for protein quantification. The level of protein denaturation was determined by precipitation of the aggregated and unfolded proteins at pH 4.6 (adjusted by 0.1 M HCl) followed by centrifugation at 15,000× g for 30 min to collect the supernatant [23]. Crude protein (Nx6.25) quantification in the total protein solution, the soluble fraction and the native fraction was performed by the DUMAS method [24], and soluble and native protein were expressed as % (*w*/*w*) of total protein. 

To gain insights in the degree of protein Maillardation, the level of carboxymethyllysine (CML), an advanced glycation end-product, was determined by UFLC/Flu method [25]. In short, protein was hydrolysed in 6M hydrochloric acid, and CML in the hydrolysate was quantified by ultra-fast liquid chromoatography (UFLC) using a pre-column derivatization with o-phtaldialdehyde and fluorimetry as detection. 

### 2.2. Protein Profile Analysis 

The protein composition of both fresh and digested EH-WPC and MP-WPC was evaluated by sodium dodecyl sulphate -polyacrylamide gel electrophoresis (SDS-PAGE) on NuPAGE^TM^ 4–12% bis-tris midi protein gel (ThermoFisher Scientific, Amsterdam, The Netherlands) performed as previously reported [26]. Equal levels of protein per sample were loaded, as determined by bicinchoninic acid (BCA) reaction [27]. Analysis was performed under reducing (with 2-mercaptoethanol (2ME)), and also under non-reducing conditions, to evaluate the protein composition of aggregates held together by disulfide bridges. Proteins and peptides were identified using the PageRuler^TM^ plus protein ladder (10–250 kDa, #26619, ThermoFisher Scientific, Amsterdam, the Netherlands), and included bovine serum albumin (BSA) 69 kDa, immunoglobulin G heavy chain polypeptide (IgG HC) 53 kDa, β-casein (βCAS) 24 kDa, β-lactoglobulin (βLG) 18 kDa and α-lactalbumin 14 kDa. Gels were scanned using Gel Doc Universal Hood II (Bio-rad, Hercules, CA, USA) and densitometric analysis was performed in Quantity One version 4.6.9 (Bio-Rad, Hercules, CA, USA). Each peak intensity displayed on the densitogram corresponds to the amount of soluble proteins that migrated through the gel to their specific protein size. Only densitograms from samples run on the same gel were compared.

### 2.3. Piglet Study

The experimental set-up of the piglet study is shown in Appendix A, as was reported previously [28]. Preterm and near-term pigs (Danish Landrace x Large White x Duroc) were delivered from sows by caesarean section at 90% gestation (106 days of gestation, *n* = 34, 2 L) and 96% gestation (113 days of gestation, *n* = 18, 1 litter) respectively. Piglets were transferred to the intensive care unit and housed individually in heated incubators with air and oxygen supply. Body temperature was recorded frequently for the first 12 h and subsequently daily, together with a daily measurement of body weight. Surgical preparation with an orogastric feeding tube and an arterial catheter for parenteral nutrition (PN) and passive immunization took place as previously described [29,30].

Piglets (males and females) from each litter were block randomized according to birthweight into two groups of enteral diets:
(1)EH-WPC group.(2)MP-WPC group.


During the study, investigators were all blinded for type of diet. The Danish National Committee on Animal Experimentation approved all procedures, which is in accordance with the EU Directive 2010/63/EU Article 23.2 and the Danish executive order no 2014-15-0201-00418.

Piglets received minimal enteral nutrition (EN) for 5 days based on EH-WPC or MP-WPC, via an orogastric tube (6F, Portex, Kent, UK). Each formula consisted of 80 g/L WPC, 50 g/L pepdite (infant milk formula containing non-milk derived low molecular weight peptides, essential amino acids, carbohydrates, fats, vitamins, minerals and trace elements), 50 g/L liquigen (medium-chain fatty acids) and 30 g/L calogen (long-chain fatty acids) (all obtained from Nutricia advanced medical nutrition). Macronutrient composition of the formulas was as follows: 3629 kJ/L energy, 59 g/L protein, 52 g/L fat, 39 g/L carbohydrate (of which 21 g/L maltodextrin, 2.7 g/L maltotriose and 1.8 g/L maltose and 16 g/L lactose).

During the study period, piglets received the enteral nutrition via the orogastric tube in increasing dose as indicated in Appendix A, with additional continuous parenteral nutrition support (Kabiven, Fresenius Kabi, Bad Homburg, Germany) through an umbilical catheter. One hour prior to euthanasia on day 5, the pigs received a last bolus EN of 15 mL/kg bodyweight. Additional enteral boluses via the orogastric tube included galactose on day 3, lactose and X-ray contrast fluid (Iodixnol, Visipaque, GE Healthcare, Brøndby, Denmark) on day 4 and lactulose/mannitol (Sigma, St Louis, MO, USA) at day 5, as indicated in Appendix A.

### 2.4. Feeding Tolerance in Piglets

Piglets were monitored very frequently throughout the day and night by caretakers and veterinarians for clinical symptoms of feeding intolerance, vomiting, abdominal distention, hemorrhagic diarrhea, respiratory distress and/or other general signs to monitor animal welfare during the entire study. Gastric residuals were measured and sampled to investigate the value of gastric residual as a marker of feeding tolerance. Aspirates were taken three times a day, prior to EN feeding (i.e., 3 h after the previous bolus). One mL of air was put into the feeding tube, after which gastric content was pulled back up until vacuum was reached. The total volume of gastric aspirate was measured and, when possible, 1 mL of aspirate was collected and stored at −80 °C. Remaining gastric aspirate was returned into the stomach gently via the tube. Fecal assessment was performed twice a day based on stool frequency and stool consistency, according to Appendix A. Both clinical and fecal assessment were performed by personnel blinded for the diet.

### 2.5. Galactose, Lactose and Lactulose/Mannitol Test

On day 3 in the morning, piglets received an enteral bolus of 15 mL/kg of 5% galactose. On day 4 in the morning, piglets received an enteral bolus of 15 mL/kg of 10% lactose. We drew 1 mL of blood from the umbilical catheter at 0 and 20 min after administration of the bolus and the increase in plasma galactose was determined for both tests. Plasma galactose was determined spectrophotometrically with the conversion of D-galactose into D-galacturonic acid by galactose dehydrogenase, as described previously [31].

On day 5, piglets received an enteral bolus of 15 mL/kg containing 5% lactulose and 5% mannitol 3 h prior to tissue collection. A urine sample was collected by intra-abdominal cystocentesis during autopsy to determine the urinary lactulose/mannitol ratio. Urinary lactulose and mannitol concentrations were determined spectrophotometrically, as described previously [31].

### 2.6. Gastro-Intestinal (GI) Transit by X-ray Analysis

On day 4 in the evening, gut transit time was assessed by X-ray photography after oral intake of a contrast solution, as described previously [32]. Each piglet received an enteral bolus of 4 mL/kg contrast fluid (4 mL/kg, Idoixnol, Visipaque^®^, GE Healthcare, Brøndby, Denmark) after 2 h of fasting, to mimic clinical practices. X-ray images of the GI tract were taken using Mobilett XP Hybrid (Siemens, Berlin, Germany) at 20 min, 1, 2 and 4 h and subsequent every second hour till the contrast fluid was cleared from the GI tract or maximally 14 h, which ever came first. The piglets were placed on their backs while the X-ray was taken and returned to their home cage between screenings. During the X-ray examination, piglets received enteral nutrition according to the regular feeding schedule (i.e., every 3 h). X-ray images were interpreted by both a neonatologist and a radiologist, blinded to the types of diet. For analysis, time of contrast to be cleared from the stomach (StEmpty), to be cleared from the small intestine (SIEmpty), to first appear at caecum (ToCaecum) and to first appear in the rectum (ToRectum) were recorded.

### 2.7. Sample Collection

Piglets were euthanized on day 5 for sample collection, or when clinical symptoms of feeding intolerance, vomiting, abdominal distention, haemorrhagic diarrhea and/or respiratory distress appeared during the study and the humane endpoint was reached. The piglets were first anaesthetized with an intramuscular injection of Zoletil mix (0.1 mL/kg, Virbac, Kolding, Denmark) and subsequently 5 mL of 20% pentobarbital (Euthanimal, Scanvet, Denmark) was injected intracardially to euthanize the piglet. The weight of the full and empty stomach was recorded to determine the volume of the gastric content, and samples of gastric content were collected and stored at −80 °C until further analysis. The small intestine was divided equally in three regions (proximal, middle and distal), weight of the content was measured, content samples were collected and stored at −80 °C until further analysis. In addition, pieces of small intestinal tissue were fixed in paraformaldehyde or snap frozen and stored at −80 °C for further analysis.

### 2.8. Analysis of Digesta

To measure pH levels in the gastric content, 0.5 mL of gastric content was diluted in 1 mL of dH_2_O, vortexed and subsequently pH was recorded with the SensION+PH3 meter (HACH).

The gastric content and the in vitro digesta obtained were poured over sequentially placed analytical sieves, with a mesh width of 2 mm, 1 mm and 0.25 mm (Retsch, VWR, Amsterdam, The Netherlands). Coagulates were separated according to their particle diameter in four fractions: larger than 2 mm (D > 2 mm), between 1 and 2 mm (2 > D > 1 mm), between 0.25 and 1 mm (1 > D > 0.25 mm) and the permeate which is smaller than 0.25 mm (D < 0.25 mm). After 30 min, wet weight of the sieves and permeate was recorded to determine the wet weight of the fractions. Protein concentration in the separated fractions was measured by BCA [27]. Sample preparation for the BCA assay included a predilution step 1:10 in Dulbecco’s phosphate-buffered saline with 2% SDS, then a second dilution 1:10 in H_2_O.

### 2.9. Enzyme Activity Assay

Lactase-phlorizin hydrolase (EC 3.2.1.23), aminopeptidase A (EC 3.4.11.7) and aminopeptidase N (EC 3.4.11.2) activity was determined in small intestinal tissue homogenates as described previously [33,34]. Conversion of lactose, glu-p-nitroanilide or l-ala-p-nitroanilide (all from Sigma-Aldrich, Zoetermeer, The Netherlands) was determined spectrophotometrically and corrected for total amount of protein in the sample, as determined by BCA reaction [27]. Enzyme activity values are expressed in units, with one unit equalling the conversion of one µM substrate/min.

### 2.10. Bacterial Staining

Paraformaldehyde-fixed proximal and distal small intestinal tissue was evaluated for the presence of adherent bacteria by fluorescence in-situ hybridisation (FISH). Slides were deparaffinised and rehydrated, after which they were incubated in a hybridisation buffer (20 mM Tris-HCl pH 7.4, 0.9 M NaCl, 0.1% SDS, all from Sigma-Aldrich, Zoetermeer, The Netherlands) for 40 min at 50 °C. Next, slides were incubated overnight at 50 °C with a 1000 ng/ul probe in a hybridisation buffer. The probe used in this study was the EUB338 Cy3 double labelled probe (EUROGENTEC, Liège, Belgium), with sequence gct-gcc-tcc-cgt-agg-agt. After washing for 20 min at 50 °C with washing buffer (20 mM Tris-HCl pH 7.4, 0.9 M NaCl), slides were mounted with ProLong^TM^ Gold antifade reagent with 4’,6-diamidino-2-fenylindool (Invitrogen, Thermo Fisher Scientific, Lelystad, The Netherlands). Images were obtained on a Leica CTR 6000 and relative staining intensity of 5 representative images per piglet were quantified using ImageJ (version 1.52a, National Institutes of Health, Bethesda, MD, USA).

### 2.11. In Vitro Digestion

Digestion of the EH-WPC and MP-WPC was simulated in vitro in the semi-dynamic digestion model (SIM) described previously [35], with minor adaptations to mimic preterm infant digestion [36]. In detail, gastric digestion was simulated over a period of 120 min at 37 °C in multi fermenter fed-batch bioreactors (Dasgip AG, Jülich, Germany). Bioreactors were filled with a bolus of 150 mL WPC (100 g/L) and were mixed by short and gentle orbital shaking of the bioreactor every 10 min. At the start of the digestion, 25 mL of simulated saliva fluid (SSF) was added to each bioreactor. During the digestion, simulated gastric fluid (SGF) was added in a dynamic manner, with 9 mL in the first 3 min followed by a continuous addition of 22.5 mL/h. To decrease pH in time hydrochloric acid (1M, Sigma-Aldrich, Zwijndrecht, The Netherlands) was added in a pre-determined dynamic flow to result in a final pH of 4.3 after 120 min. SSF (pH 6.3) consisted of 0.1 M NaCl, 30 mM KCl (Merck, VWR International; Amsterdam, the Netherlands), 2 mM CaCl_2_.2H_2_O, 14 mM NaHCO_3_, 0.06% (*w*/*v*) α-amylase (from *Aspergillus oryzae*, A9857) (all from Sigma-Aldrich, Zwijndrecht, The Netherlands). SGF (pH 4.0) contained 50 mM NaCl, 15 mM KCl (Merck, VWR, Amsterdam, the Netherlands), 1 mM CaCl_2_.2H_2_O, 0.005% (*w*/*v*) pepsin (from porcine gastric mucosa, P7125), 0.013% (*w*/*v*) lipase (from *Rhizopus oryzae*, 80612) (all from Sigma-Aldrich, Zwijndrecht, The Netherlands).

### 2.12. Statistical Analysis

The piglet experiment was an exploratory study with a limited number of animals and, therefore, limited power, especially for the near-term piglets. Analysis was performed on *n* =14 for the preterm piglets and *n* = 8–9 for the near-term piglets. Data for the protein analysis and in vitro experiments were derived from *n* = 3 independent digestion experiments. Statistical analysis was performed using GraphPad Prism version 8 software (La Jolla, CA, USA). *p* < 0.05 was considered statistically significant. For all in vivo parameters evaluated, the effect of diet was tested only within the different gestational age groups, comparing EH-WPC with MP-WPC by a Mann–Whitney test. Data are expressed as median with interquartile range (IQR) since most data were not normally distributed (determined by D’Agostino test). For the in vitro analysis, the difference between EH-WPC and MP-WPC was determined by Student’s *t*-test. Data are expressed as mean ± standard error of the mean (SEM).

## 3. Results

### 3.1. Characterization of WPCs: Impact of Thermal Processing on Whey Protein Nativity, Carboxymethyllysine (CML) Formation and Protein Aggregation

An overview of the processing steps to obtain WPC with different levels of heat treatment is provided in Figure 1a (see also the experimental procedures). To determine the impact of heat treatment on the MP-WPC and EH-WPC, the Maillardation of proteins was evaluated by measuring the level of carboxymethyllysine (CML), an advanced glycation end-product. Compared with EH-WPC, there was a significantly lower level of CML detected in the MP-WPC (Figure 1b).

Next, protein solubility and denaturation were evaluated. Extensive heating only mildly affected protein solubility at pH 7.1, with 98% soluble protein for MP-WPC compared with 87% for EH-WPC (Figure 1c), indicating less insoluble protein aggregates in the MP-WPC. Protein denaturation was greatly affected by the extensive heat treatment, because at pH 4.6 72% soluble protein remained for MP-WPC compared with 21% for EH-WPC. As caseins and non-native whey proteins are known to precipitate at pH 4.6 [23], and as 24–28% of the protein in the WPC used in this study was β-casein (Table 1), our data reveal that the whey protein fraction in MP-WPC is close to 100% native and only 30% native in EH-WPC.

Finally, the protein composition of both EH-WPC and MP-WPC was assessed by SDS-PAGE (Figure 1d), with the corresponding peak intensities displayed in the densitogram (Figure 1e). As a result of heat treatment, disulfide bond formation can occur between two cysteine residues present in the proteins, creating aggregation of proteins. Comparing protein profiles under non-reducing (−2ME) versus reducing (+2ME) conditions provides information about the specific proteins aggregated or entrapped by protein aggregates [26]. Both BSA and IgG (HC) were aggregated, as these bands disappeared under non-reducing conditions for both EH-WPC and MP-WPC. BSA consists of mixed disulfides that can form polymers, which react very rapidly to give aggregates upon heat treatment, even after only mild pasteurization [37]. IgG heavy chains are connected by disulfide bridges to form dimers in normal conformation [38]. The effect of extensive heat treatment was most striking for α-lactalbumin and to a smaller extent also for β-lactoglobulin, with a decrease in band intensity under non-reducing conditions for EH-WPC only. These data imply that α-lactalbumin and β-lactoglobulin in the MP-WPC are present freely without a covalent link to other proteins, but upon extensive heating become a covalently linked part of protein aggregates/polymers. Taken together, WPC characterization showed Maillardation of proteins, reduced protein nativity and increased protein aggregation upon extensive heat treatment.

### 3.2. Piglet Study: Clinical Symptoms and Adverse Events

Several cases of adverse events were observed in the preterm piglets during the five-day dietary intervention, including feeding intolerance, abdominal distention, hemorrhagic diarrhea and/or respiratory distress. The incidence of these adverse events was equal for both diets (*n* = 3 for MP-WPC and *n* = 3 for EH-WPC) and was restricted to one preterm litter. Because these piglets were taken out of the study before day 5 without receiving a last bolus of enteral nutrition, most GI tolerance parameters could not be evaluated and, therefore, these piglets were excluded from all analysis. No diet-related clinical adverse events were reported for the near-term piglets during the study. One near-term piglet had to be euthanized according the human end-point due to catheter-related sepsis and was, therefore, also excluded from analysis. Parameters of growth, intestinal integrity and innate defense evaluated in these piglets are reported elsewhere [28]. Postnatal growth was not significant different between MP-WPC and EH-WPC based diet groups. Independent of diet, preterm piglets did show lower weight gain than near-term piglets [28].

Fecal scores, based on stool frequency and consistency (Appendix A), showed minimal to no feces on day 1, 2 and 3 for any of the preterm or near-term piglets, which can be expected with minimal volumes of enteral nutrition. There was an extensive increase in fecal scores on day 4 and 5 (Figure 2a–c). The overall increase from 9 a.m. to 6 p.m. at day 4 was potentially induced by the oral lactose challenge, while the increase from day 4 to 5 might be the result of the X-ray contrast fluid. With the increasing fecal scores, there was a strong trend towards lower fecal scores for preterm piglets receiving MP-WPC compared with EH-WPC on day 4 at 6 p.m. (Figure 2b) and day 5 at 9 a.m. (Figure 2c), suggesting improved GI tolerance in preterm piglets fed MP-WPC.

### 3.3. Piglet Study: GI Tolerance

An important indicator of feeding intolerance in preterm infants used in the clinic is the presence of gastric residuals, which is assessed by taking gastric aspirates. During this study, remarkable differences in gastric residual volumes 3 h after an enteral bolus were observed. In general, the volume of gastric residuals decreased over the days, with detectable gastric residuals in most piglets at day 1 (Figure 2d) but undetectable gastric residuals in most piglets at day 4 (Figure 2g). In preterm piglets, the volume of gastric residuals was higher in the piglets receiving MP-WPC compared with EH-WPC on all days (Figure 2d–g), suggesting that either the MP-WPC receiving piglets had more diet left in the stomach, or that the EH-WPC residuals could not be aspirated via the small diameter of the tube. In near-term piglets, aspirate volumes were low and not significantly different between MP-WPC and EH-WPC on all days (Figure 2d–g). When assessing stomach emptying (StEmpty) of contrast fluid by X-ray on day 4, there were no differences between MP-WPC and EH-WPC in either preterm (Figure 2h) or near-term piglets (Figure 2i).

### 3.4. Piglet Study: Digestive and Absorptive Capacity and Intestinal Permeability

To assess the digestive and absorptive capacity of the small intestine, the galactose and lactose challenge tests were performed at day 3 and day 4 respectively. Plasma galactose levels increased similarly for both diets 20 min after an oral bolus of galactose (Appendix A), while there was no detectable increase in plasma galactose levels 20 min after an oral bolus of lactose. The lactulose/mannitol ratios measured in the urine at day 5, as an indicator of small intestinal permeability, was lower in near-term piglets receiving MP-WPC compared with EH-WPC (Appendix A). Due to incomplete urine collection, lactulose/mannitol ratio could not be measured accurately for the preterm piglets. Brush border enzyme activity levels were at similar levels for MP-WPC and EH-WPC groups (Appendix A).

### 3.5. Gastric Content: pH and Formation of Coagulates In Vivo

To decipher whether the MP-WPC was emptied slower from the stomach or whether the EH-WPC residual was too thick for aspiration via the gastric tube, the volumes of the gastric contents were measured during necropsy at day 5, 1 h after a 15 mL/kg bolus of EN. In contrast to the gastric aspirates, the volume of gastric content was significantly lower in the piglets receiving MP-WPC compared with EH-WPC for both preterm and near-term piglets (Figure 3a). Moreover, the volume of the gastric content was for most piglets receiving EH-WPC higher than the volume of the last bolus administered, suggesting an additional effect of previous boluses. The gastric content volumes did not correlate to gastro-enterocolitis score (Appendix A). After collection of the content, the relative stomach weight of piglets receiving MP-WPC was lower compared with piglets receiving EH-WPC (Figure 3b), suggesting less inflammation and/or edema.

The pH levels of the gastric content tended to be higher in the MP-WPC compared with EH-WPC for preterm piglets (Figure 3c), while this difference was less pronounced for the near-term piglets. There was a significant strong inverse correlation between gastric content volume and pH level for the preterm piglets (Figure 3d, Pearson’s r −0.49 *p* = 0.02) and a similar trend in the near-term piglets (Figure 3e, Pearson’s r −0.47 *p* = 0.06).

The gastric content of 6 near-term piglets (*n* = 3 for each diet) were further analyzed to determine the formation and size of coagulates. The coagulates in the gastric content of piglets fed MP-WPC were smaller in weight and size compared with the coagulates in the gastric content of piglets fed EH-WPC (Figure 3f). The volume of the permeate (D < 0.25 mm) was equal for both diets (Figure 3f).

Taken together, although the volumes of gastric aspirates were significantly higher, the gastric content was significantly lower in preterm piglets fed MP-WPC compared with EH-WPC, indicating that the gastric EH-WPC residuals were too thick for complete aspiration and thereby led to an underestimation of the real gastric residual volume. Near-term piglets showed similar lower gastric content for MP-WPC, with a lower volume of coagulates in the gastric content present compared with EH-WPC.

### 3.6. GI Transit and Mucosal Bacterial Adherence

Coagulation of proteins can delay gastric emptying and impact GI transit; therefore, the small intestine (SI) and colon content was determined (Figure 4a–d). The volume of the distal SI content (Figure 4c) and colon content (Figure 4d) were significantly higher in the preterm piglets fed MP-WPC compared with those fed EH-WPC based formula. In the near-term piglets, this effect was not observed. Evaluation of the GI transit time of the contrast fluid by X-ray revealed that there was no difference in time to empty the small intestine (SIEmpty) between MP-WPC and EH-WPC for both preterm (Figure 4e) and near-term piglets (Figure 4f). In addition, there was no difference in time of the contrast fluid to first appear in the cecum (Appendix A). Colonic transit time could not be determined, because the contrast fluid had not yet reached the rectum in a high percentage of the preterm piglets after 12 h (Appendix A). In most near-term piglets, the contrast fluid did reach the rectum, corresponding to the higher incidence of diarrhea observed in these piglets.

Next, the presence and adherence of bacteria to the intestinal epithelium was evaluated. Epithelial-adhering bacteria were detected by FISH with a probe for eubacteria (Figure 4g). Adherence of bacteria to the proximal SI epithelium in the preterm piglets was higher than in the near-term piglets. In the distal SI, bacteria were identified at the villi of all piglets. Quantification revealed significant lower EUBAC staining intensity in preterm piglets exposed to MP-WPC diet compared with EH-WPC in the proximal SI (Figure 4h), while this effect was absent in the distal small intestinal tissue (Figure 4i).

Taken together, preterm piglets fed MP-WPC based formula had less gastric residuals and more intestinal content, especially towards the distal region of the GI tract, but the transit time of contrast fluid was similar. Bacterial attachment results imply that a diet based on MP-WPC can decrease mucosal bacterial adherence in the proximal SI of preterm piglets compared with a diet based on EH-WPC. For near-term piglets, these effects were not observed.

### 3.7. In Vitro Digestion and Coagulation

To investigate the differences in coagulation between EH-WPC and MP-WPC in more detail, gastric digestion was simulated in vitro in the SIM model using digestion conditions specific for preterm infants. After 2 h of in vitro digestion, the digesta were poured over sequentially placed sieves separating them based on particle size. Coagulates >1 mm were nearly undetectable in the gastric digesta of MP-WPC nor EH-WPC (Figure 5a). There were significantly fewer particles with a size between 0.25 and 1 mm for the MP-WPC compared with EH-WPC, while the weight of the permeate (i.e., D < 0.25 mm) was significantly larger in the MP-WPC compared with EH-WPC (Figure 5a).

It was impossible to sample coagulates with size >2 mm and 2–1 mm for MP-WPC due to minimal digesta retained on the sieves. However, protein concentrations of the coagulated fraction 1 > D > 0.25 mm tended to be lower for MP-WPC compared than EH-WPC (Figure 5b). Analysis of the protein composition in the different fractions analyzed by SDS-PAGE identified similar patterns between MP-WPC and EH-WPC (Figure 5c). Based on protein concentrations measured in the separated fractions and the band intensity detected by SDS-PAGE, the concentration of specific proteins was calculated. Coagulates that were formed had comparable protein composition for both EH-WPC and MP-WPC (Appendix A). As expected, the relative abundance of β-casein was higher in the coagulate compared with the permeate, since this protein precipitates at low pH [23]. As a consequence, the relative abundance of β-lactoglobulin, the most abundant whey protein, was lower in the coagulate compared with the permeate.

Comparing the coagulates from MP-WPC with the coagulates from EH-WPC revealed significant differences in protein composition. The concentration of β-lactoglobulin in the coagulates with particle size 1 > D > 0.25 mm was significantly lower in digested MP-WPC compared with digested EH-WPC (Figure 5d), with the same tendency for α-lactalbumin. Bovine serum albumin, immunoglobulin G (HC) and β-casein were present in these coagulates in similar concentrations for both MP-WPC and EH-WPC. In the permeate (D < 0.25 mm), there were no differences in protein concentration detected between MP-WPC and EH-WPC for any of the specific proteins (Appendix A).

In summary, MP-WPC shows a lower level of protein coagulation after in vitro simulated digestion compared with EH-WPC, with lower levels of β-lactoglobulin and α-lactalbumin in the coagulates.

## 4. Discussion

In this study, the gastro-intestinal tolerance of mildly pasteurized WPC (MP-WPC) was compared with more extensive heated WPC (EH-WPC) in a piglet model highly sensitive to dietary interventions. Five days of minimal enteral nutrition based on MP-WPC reduced gastric residual volume and increased intestinal content in preterm piglets. In addition, preterm piglets receiving MP-WPC showed less diarrhea and reduced mucosal bacterial adherence. Protein analysis revealed lower levels of protein aggregate and less coagulation for MP-WPC, suggesting that minimal heat treatment of WPC promotes gastro-intestinal tolerance in immature piglets.

The volume of gastric residuals aspirated via the feeding tube during the piglet study was higher for MP-WPC compared with EH-WPC, while opposite results were found for the gastric content/residual volume present at necropsy. It is likely that the coagulated gastric content could not be aspirated via the feeding tube and, thereby, impaired accurate evaluation of the gastric residuals by aspiration. This might also explain the low clinical value of gastric residuals to predict necrotizing enterocolitis in preterm infants [39]. The gastric content volumes in this study did not correlate with the gastroenterocolitis scores, excluding a potential negative feedback regulation on gastric emptying as a consequence of intestinal damage.

The remarkably higher gastric content in piglets receiving EH-WPC compared with piglets receiving MP-WPC suggests delayed and/or incomplete gastric emptying in piglets fed EH-WPC. Previous findings that piglets receiving infant milk formula have reduced gastric emptying compared with bovine colostrum suggest that like colostrum, MP-WPC improves gastric emptying [40]. The difference in gastric emptying between MP-WPC and EH-WPC was most likely caused by the differences in coagulation and formed coagulates which were observed in both the in vivo and in vitro setting. In vivo, there was an important additional effect of gastric residuals that accumulated over time due to seriated bolus feeding. This may have accounted for the large coagulates (D > 2 mm) observed in the gastric content in vivo while these were (nearly) absent in the in vitro setting. Indeed, in vitro digestion pilot experiments with repeated digestion of coagulates with addition of a second protein bolus also resulted in larger coagulates (data not shown). In addition, in vivo there are several feedback mechanisms of digestive hormones, neural interactions and absorption that affect digestion, gastric emptying and GI transit [41], which are lacking in the in vitro setting.

Characterization of the MP-WPC and EH-WPC revealed a strong effect of extensive heat treatment on protein nativity. Unlike caseins, whey proteins are strongly affected by thermal processing [42]. Whey proteins are easily denatured, i.e., protein structures are unfolded, disulfide bridges are broken, and upon cooling aggregates and/or polymers are formed, especially between β-lactoglobulin and α-lactalbumin [8,42]. In this study, both β-lactoglobulin and α-lactalbumin appeared to aggregate upon extensive heat-treatment. Subsequent digestion and coagulation studies revealed that MP-WPC coagulated less compared with EH-WPC and that the specific proteins within the coagulates were a mixture of all proteins present in the WPC. Typically, casein coagulates are under gastric conditions, while whey proteins remain liquid and encounter no delay in gastric emptying [43]. In our study, however, it could be that extensively heated whey proteins formed a gel induced by the gastric acidity and, thereby, contributed to protein coagulation. This would be in line with a previous study showing that coagulation of WPC can be induced by a low pH [44]. Furthermore, although the coagulates consisted of a mixture of all proteins present within the diet, it appeared that the coagulated fraction (1 > D > 0.25 mm) of MP-WPC contained less β-lactoglobulin and α-lactalbumin compared with the coagulated fraction of EH-WPC. This is most likely a result of the lower protein aggregation observed for MP-WPC and further underlines the relation between WPC heating and formation of gastric coagulates. In addition, the lower pH in the gastric content of EH-WPC fed piglets might have induced a higher degree of protein coagulation and larger coagulates, that cannot pass the pylorus and, therefore, accumulate in the stomach. The cause of the lower pH remains unknown however, as both MP-WPC and EH-WPC had a pH of 7.1 when administered. The increased gastric residual volume might have resulted in mechanical stimulation (distention) of receptors, which subsequently increase HCl secretion in the stomach. The lower pH subsequently can result in even more coagulation of the proteins. This phenomena however is only described in adults [45], and not yet studied in (preterm) infants.

Coagulation of milk in the stomach is a physiological process to facilitate proteolysis of the casein proteins. Two potential effects of increased protein coagulation during digestion are reduced gastric emptying and changed digestion and absorption of proteins, leading to changed bio-availability and nutritional value of proteins [14]. In this study, there was a significantly higher level of gastric emptying and more intestinal content in the distal SI and colon of piglets receiving MP-WPC compared with EH-WPC, while there were no differences measured in absorption capacity based on oral challenge tests and enzyme activity levels. These results correspond to the higher volumes of permeate measured after in vitro digestion of MP-WPC, with particle size that can pass the pylorus (i.e., D < 0.25 mm). The contrast fluid was emptied from the stomach in similar rate for both diets, indicating that the differences in gastric emptying observed were mainly caused by the coagulation and not due to differences in motility.

Previously, it has been shown that native whey proteins are more resistant to hydrolysis in the stomach than denatured whey proteins and, therefore, intact whey proteins can reach the SI to exert their functions while denatured whey proteins are more hydrolyzed in the gastric phase [46]. By limiting heat load during processing, the denaturation of whey proteins was prevented, thereby maintaining protein bioactivity and limiting gastric digestion of these proteins. In turn, this leads to (more) bioactive whey proteins reaching the SI. Despite not showing any differences in absorption between the diets, the bolus of galactose (Day 3), lactose (Day 4) and lac/man (Day 5) did challenge the GI tract. For example, the increase in fecal scores at day 4 from 9 a.m. to 6 p.m. was likely induced by the oral bolus of lactose and the additional increase in diarrhea at day 5 was likely caused by the contrast fluid administration. Further research is needed to reveal the impact of the MP-WPC and EH-WPC on GI tolerance without these oral challenges.

Piglets are fermenters and, thus, rely on bacteria present in the intestine for metabolism [41]. Preterm piglets cannot digest maltodextrins efficiently, which can result in bacterial overgrowth, increased fermentation and potentially induce diarrhea and/or mucosal inflammation [47,48,49]. In contrast to piglets, preterm infants seem to be able to digest maltodextrins more efficiently, however compared with term infants the capacity might still be limited. In this study, we demonstrated a potential protective effect of MP-WPC by limiting bacterial adherence compared with EH-WPC. It is described that formula feeding in preterm piglets is associated with an enhanced colonization by mucosal adhering bacteria relative to feeding with sow’s colostrum [20,50]. In addition, there is a higher risk for gut colonization by pathogenic bacteria in formula-fed neonatal piglets compared with piglets fed sow’s milk [51]. Data presented here suggest that MP-WPC might have protective effects by limiting bacterial adherence as has been described for sow’s colostrum. Of note, no antibiotics were administered in our study. The protective effect observed might be caused by direct effects of the MP-WPC, which contained more native whey proteins and, therefore, has potentially higher anti-bacterial capacity. In addition, the MP-WPC might have indirectly decreased the bacterial adherence by influencing the colonization of specific microbes thereby changing microbial composition and homeostasis. Nevertheless, further studies are needed to clarify the exact mechanisms by which MP-WPC limits bacterial adherence to the intestinal epithelium.

In summary, mildly pasteurized WPC improves gastro-intestinal tolerance in immature piglets compared with extensively heated WPC. These findings will contribute to the optimization of infant milk formula for (preterm) infants, reducing the feeding intolerance that is currently still frequently observed and supporting a healthy intestinal development. However, it should be taken into account that the current study only included one near-term piglet litter and further studies are warranted regarding differences between gestational ages.

## Figures and Tables

**Figure 1 nutrients-12-03391-f001:**
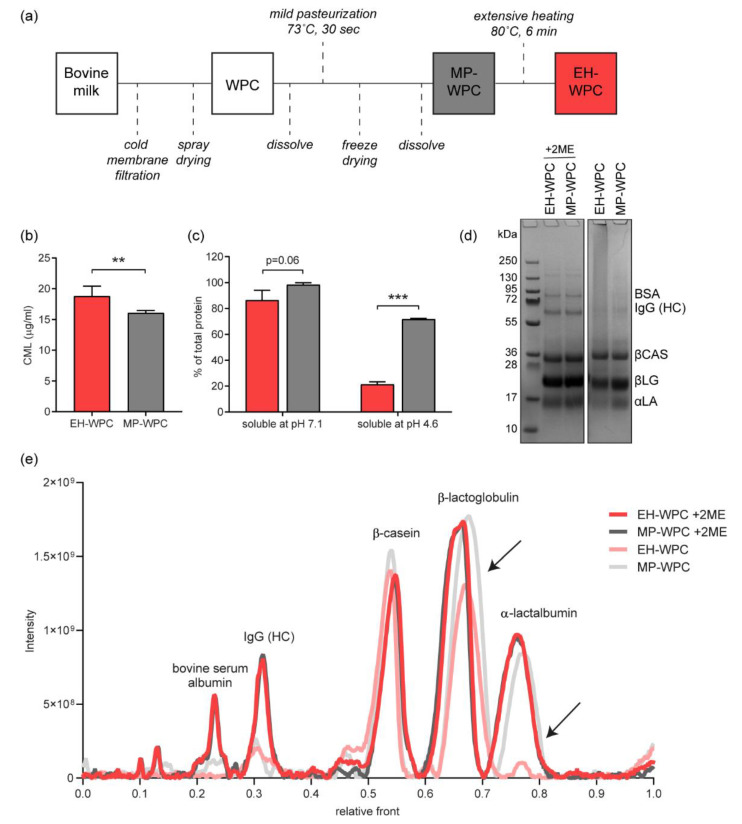
Whey protein processing. (**a**) Whey protein concentrate (WPC) was obtained from raw bovine milk by cold membrane filtration. The WPC was first mildly pasteurized for 30 s at 37 °C, after which a part of the WPC was extensively heat-treated for 6 min at 80 °C. (**b**) The level of carboxymethyllysine (CML, µg/mL) measured by ultra-fast liquid chromatography/fluorescence UFLC/Flu) in both mildly pasteurized (MP-WPC) and more extensively heated WPC (EH-WPC). (**c**) Soluble protein (% of total WPC protein) at pH 7.1 and pH 4.6, determined by the DUMAS method. (**d**) Representative sodium dodecyl sulphate-polyacrylamide gel electrophoresis (SDS-PAGE) gel of EH-WPC and MP-WPC at pH 7.1 under reducing (+2ME) and non-reducing conditions, with (**e**) corresponding densitometric analysis. Black arrows indicate decrease in peak intensity for EH-WPC specifically. *n* = 3, ** *p* < 0.05, *** *p* < 0.01 based on t-test between EH-WPC and MP-WPC. BSA = bovine serum albumin, IgG (HC) = immunoglobulin G (heavy chain), βCAS = β-casein, βLG = β-lactoglobulin, αLA = α-lactalbumin. kDa = kilodaltons, 2ME = 2-mercaptoethanol.

**Figure 2 nutrients-12-03391-f002:**
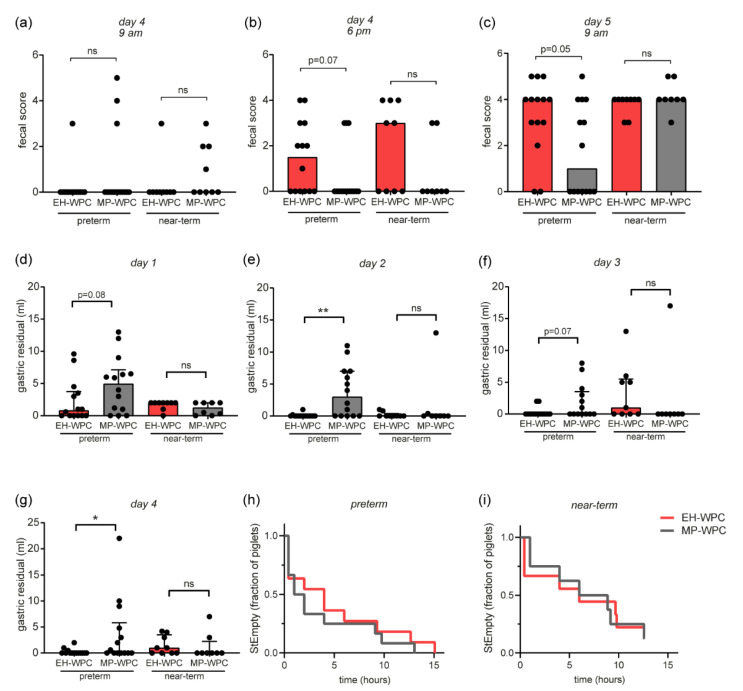
Clinical observations piglet study. (**a**–**c**) Fecal scores of preterm and near-term piglets at (**a**) day 4 at 9 am, (**b**) day 4 at 6 p.m. and (**c**) day 5 at 9 am. (**d**–**g**) Gastric aspirates were taken 3 h after a bolus of enteral nutrition (EN) and volume (mL) was measured at (**d**) day 1, (**e**) day 2, (**f**) day 3 and (**g**) day 4. (**h**,**i**) Fraction of piglets that not cleared the contrast fluid from the stomach over time (**h**), determined by X-ray analysis, for (**h**) preterm and (**i**) near-term piglets. *n* = 14 for preterm piglets, *n* = 8–9 for near-term piglets. * *p* < 0.05, ** *p* < 0.01, ns = non-significant based on t-test between EH-WPC and MP-WPC.

**Figure 3 nutrients-12-03391-f003:**
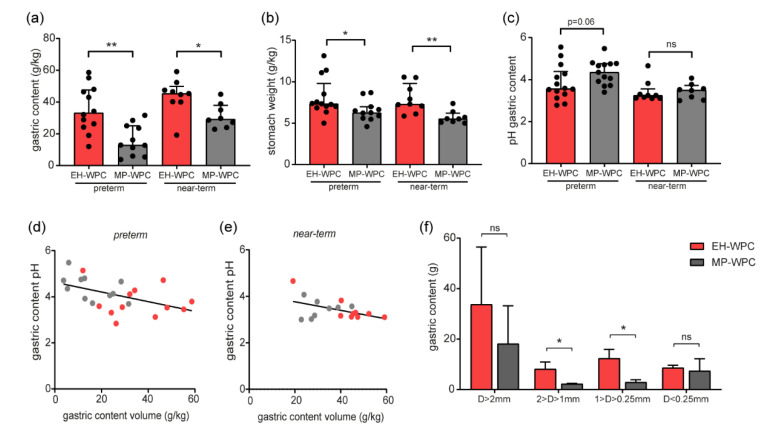
Digestion and coagulation in vivo. (**a**) gastric content weight and (**b**) relative weight of the empty stomach (g/kg bodyweight) were measured at day 5 during necropsy, 1 h after the last bolus of EN, with (**c**) pH levels of the gastric content. (**d**,**e**) correlation between gastric content volume and pH levels for (**d**) preterm and (**e**) near-term piglets. *n* = 14 for preterm piglets, *n* = 8–9 for near-term piglets. * *p* < 0.05, ** *p* < 0.01, ns = non-significant based on Mann–Whitney test between EH-WPC and MP-WPC. (**f**) Gastric content of 6 near-term piglets (*n* = 3 per diet) was separated based on particle size and weight of separate fractions (g) was recorded. * *p* < 0.05, ** *p* < 0.01, ns = non-significant based on t-test between EH-WPC and MP-WPC.

**Figure 4 nutrients-12-03391-f004:**
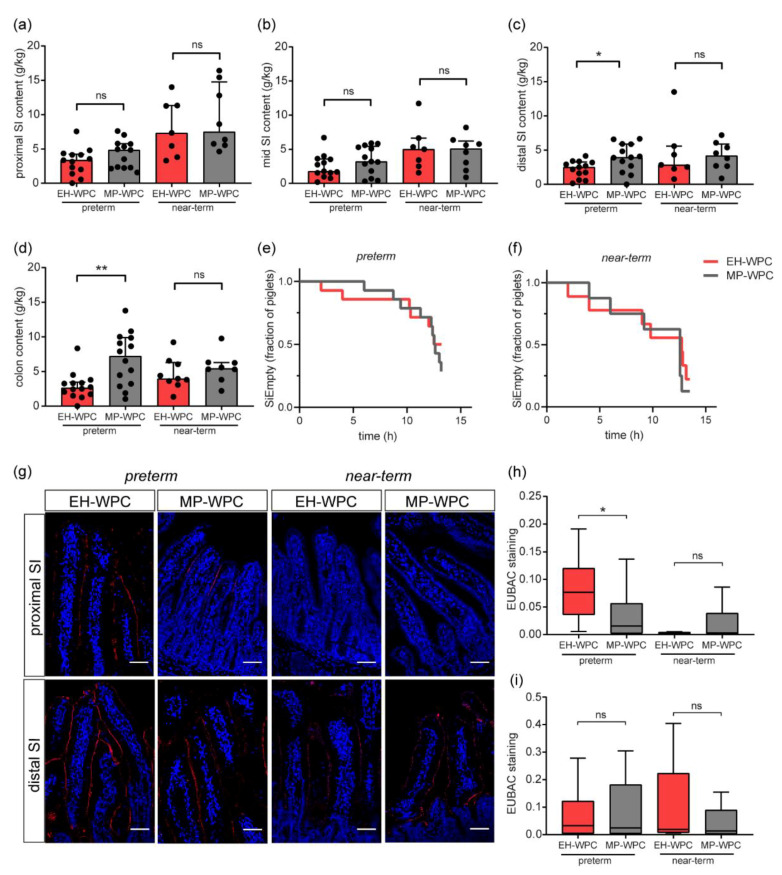
GI transit and mucosal bacterial adherence. (**a**–**d**) Weight of small intestine and colon content was determined at day 5 (g/kg bodyweight), 1 h after the last bolus of EN, in separated regions (**a**) proximal small intestine, (**b**) mid small intestine, (**c**) distal small intestine and (**d**) colon. (**e**,**f**) Fraction of piglets that not cleared the contrast fluid from the small intestine over time (**h**), determined by x-ray analysis, for (**e**) preterm and (**f**) near-term piglets. (**g**) Fluorescent in-situ hybridization (FISH) detection of eukaryotic bacteria in proximal and distal small intestinal tissue slides, with quantification of staining intensity in (**h**) proximal small intestine and (**i**) distal small intestine. *n* = 14 for preterm piglets, *n* = 8–9 for near-term piglets. white scale bar equals 50 µm. * *p* < 0.05, ** *p* < 0.01, ns = non-significant based on Mann–Whitney test between EH-WPC and MP-WPC. SI = small intestine, EUBAC = eubacteria.

**Figure 5 nutrients-12-03391-f005:**
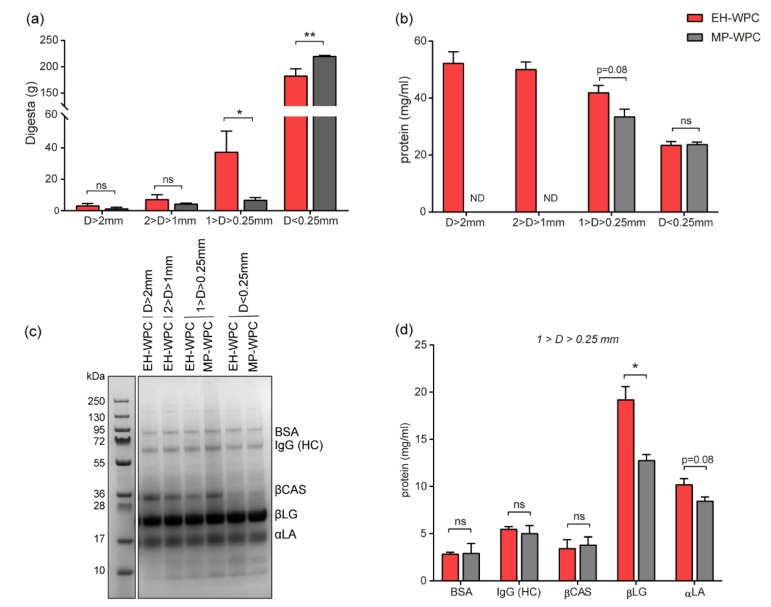
Digestion and coagulation in vitro. (**a**) In vitro digested WPC was separated based on particle size and weight of the separate fractions (g) was recorded. (**b**) Protein concentration of the separated fractions determined by BCA. (**c**) Representative SDS-PAGE gel of digested fractions of EH-WPC and MP-WPC under reducing conditions. (**d**) Protein concentrations calculated based on SDS-PAGE separation for the coagulate with a particle size 1 > D > 0.25 mm. *n* = 3, ND = not detectable, * *p* < 0.05, ** *p* < 0.01, ns = non-significant based on t-test between EH-WPC and MP-WPC. BSA = bovine serum albumin, IgG (HC) = immunoglobulin G (heavy chain), βCAS = β-casein, βLG = β-lactoglobulin, αLA = α-lactalbumin.

**Table 1 nutrients-12-03391-t001:** Protein composition of the Whey Protein Concentrate (WPC).

% of Total Protein	EH-WPC	MP-WPC
β-lactoglobulin	39%	37%
α-lactalbumin	20%	23%
β-casein	28%	24%
others	13%	16%

The percentage of specific casein and whey proteins in WPC at pH 7.1, determined by sodium dodecyl sulfate polyacrylamide gel electrophoresis (SDS-PAGE) analysis. EH-WPC = extensively heated whey protein concentrate, MP-WPC = mildly pasteurized whey protein concentrate.

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
