# Peer review of "Mildly Pasteurized Whey Protein Promotes Gut Tolerance in Immature Piglets Compared with Extensively Heated Whey Protein"

_nutrients, 2020, doi:10.3390/nu12113391_

Round 1
Reviewer 1 Report
Attached file

Author Response
This manuscript the hypothesis that mildly pasteurized formula milk will improve gastro-intestinal tolerance in immature piglets is tested, by the comparison of preterm and near term animals. The results point to a different effect with regard to the gestational age, because some parameters show different response in the two studied groups. The data are well organized and presented. Probably the discussion should be more critical and some limitations of the work could be indicated. For instance, the administration of the oral bolus of lactose and contrast fluid could be regarded as disturbing interventions that should be compared with the experiment without them.
We thank the author for taking the time to evaluate this manuscript and provide valuable feedback. We have implemented the limitations in the discussion. In line 548-552: “the bolus of galactose (day 3), lactose (day 4) and lac/man (day 5) did challenge the GI tract. For example, the increase in fecal scores at day 4 from 9 am to 6 pm was likely induced by the oral bolus of lactose and the additional increase in diarrhea at day 5 was likely caused by the contrast fluid administration. Further research is needed to reveal the impact of the MP-WPC and EH-WPC on GI tolerance without these oral challenges.”
We have implemented the other comments as indicated in detail below:
Some comments:
A graphical scheme of the animal study including the different interventions would help the reader to rapidly capture the study design
We agree with the reviewer that a graphical scheme of the piglet study is valuable to capture the study design. This was provided in the previous publication considering this study (Nutrients 2020, 12, doi: 10.3390/nu12041125), and is now also added to this manuscript as Figure S1.
Line 217. The authors should indicate how is the BCA analysis performed in fractions with coagulates.
We apologize this information was missing. Sample preparation for the BCA assay includes a predilution step 1:10 in DPBS with 2% SDS, then a second dilution 1:10 in H2O to dissolve the coagulates. We have now added this to the methods section (line 226-227).
Lines 274-278. The authors might have analyzed by SDS-PAGE the soluble fractions at pH 7.1 and 4.6 in order to compare the profiles and show in a direct way the native proteins remaining.
The samples in this study have only been analyzed by SDS-PAGE at pH 7.1, hence we are not able to display the composition of the native protein fraction. We thank the author for this valuable suggestion for future research.
Lines 466-467. This is only observed for preterm animals.
We have rephrased this in the text.
Figure 1. The number of replicates must be indicated in the case of figures (b) and (c)
We apologize if this was not clear. All in vitro analyses were performed n=3 times and this is indicated in the figure legend of figure 1 (line 314) and in section 2.12 Statistical analyses.
Figure 3 (a) The y-axis should be gastric volume (ml) and not gastric content (g/kg).
Figure 3a contains the gastric content weight in grams, which was corrected for body weight of the piglet. We apologize that the figure legend was maybe confusing, referring to gastric content volume, and therefore we have corrected this to gastric content weight (g/kg).
Figure 5 (c) This experiment was apparently performed under reducing conditions.
The reviewer is correct on this. We have added this information to the figure legend.
Reviewer 2 Report
This paper describes a study that aimed to investigate the effects of heat treatment of whey protein concentrate (mild pasteurization or extensive heating) on digestion outcomes. The study consists contains 3 stages, i.e. the development of the WPC's and the characterization of the protein profile, an in vivo study to determine the feeding tolerance of the MP-WPC and EH-WPC in pre-term and near-term piglets, and an in vivo study to further characterize the gastric digestion of the WPC’s. The authors found that MP-WPC and EH-WPC have a similar protein profile, however, the MP-WPC is better tolerated by the piglets than the EH-WPC, as indicated by less diarrhea, and a lower gastric content and smaller sized coagulates after the MP-WPC compared to EP-WPC on the necropsy day.
The study is very well designed and executed, the inclusion of pre-term as well as near term piglets generates novel data, and the results are presented elegantly. The paper is well written and was a pleasure to read.
I have a few comments for the authors’ consideration.
Main comment:
- In the aim of the study, the authors mention that they added a group of near-term pigs to the premature piglets to investigate whether the GI tolerance was influenced by the gestational age. However, with the current statistical analyses, the authors only investigate the effect of treatment within each piglet group. Was the aim established a priori? If so, should there also be statistical comparisons between the piglet groups to address this aim?
Minor comments:
- Title: the title should include the comparator (i.e. extensively heated whey protein) for clarity.
- Line 70: How does cold membrane filtration differ from traditional milk processing for IMF? Please specify in the manuscript.
- Line 70: ‘resemble’ should be changed to ‘resembles’.
- Methods: Since the WPC’s contain both whey protein and caseins, I find the term ‘whey protein concentrate’ slightly confusing.
- Line 361-363 Moreover, the volume…of previous boluses: does this sentence refer to EH-WPC only or to both WPC’s?
- Line 433: remove 1 m in ‘mmm’.
- Discussion: Is there any reason for concern around food safety of the milk with the reduced heating time of MP-WPC compared to EH-WPC or conventional milk processing? If so, it would be good to address this in the discussion.
- I think the table S1 contains essential information that would be best to include in the manuscript – would the authors consider this?
Author Response
This paper describes a study that aimed to investigate the effects of heat treatment of whey protein concentrate (mild pasteurization or extensive heating) on digestion outcomes. The study consists contains 3 stages, i.e. the development of the WPC's and the characterization of the protein profile, an in vivo study to determine the feeding tolerance of the MP-WPC and EH-WPC in pre-term and near-term piglets, and an in vivo study to further characterize the gastric digestion of the WPC’s. The authors found that MP-WPC and EH-WPC have a similar protein profile, however, the MP-WPC is better tolerated by the piglets than the EH-WPC, as indicated by less diarrhea, and a lower gastric content and smaller sized coagulates after the MP-WPC compared to EP-WPC on the necropsy day.
The study is very well designed and executed, the inclusion of pre-term as well as near term piglets generates novel data, and the results are presented elegantly. The paper is well written and was a pleasure to read.
We thank the author for taking the time to evaluate this manuscript, for the kind words and for providing valuable feedback. We have implemented the comments, as indicated in detail below:
I have a few comments for the authors’ consideration.
Main comment:
In the aim of the study, the authors mention that they added a group of near-term pigs to the premature piglets to investigate whether the GI tolerance was influenced by the gestational age. However, with the current statistical analyses, the authors only investigate the effect of treatment within each piglet group. Was the aim established a priori? If so, should there also be statistical comparisons between the piglet groups to address this aim?
We agree with your comment, the aim of the study was stated indistinctly. The primary aim of this study was to investigate the impact of a MP-WPC based formula and EH-WPC based formula on gut tolerance of preterm piglets in more detail. Secondly, we had the opportunity to also explore if the observed differences between the two diets was similar for near term piglets. We have rephrased the aim for clarity (line 87-94).
Minor comments:
1. Title: the title should include the comparator (i.e. extensively heated whey protein) for clarity.
We agree with the reviewer that it is more clear to have the comparator in the title and have therefore added this.
2. Line 70: How does cold membrane filtration differ from traditional milk processing for IMF? Please specify in the manuscript.
Traditional milk processing for IMF involves rennet or acid precipitation to separate the soluble whey protein fraction from the precipitated casein. This is a completely different method than the membrane filtration used here, where a WPC is obtained that not only contains an improved casein to whey ratio, but also a casein profile resembling human milk. We have explained this a bit more extensively in the introduction now.
3. Line 70: ‘resemble’ should be changed to ‘resembles’.
Thank you for noticing, this is now corrected.
4. Methods: Since the WPC’s contain both whey protein and caseins, I find the term ‘whey protein concentrate’ slightly confusing.
We agree with the reviewer; however this is the general and correct term used. WPC is a protein mixture that has been concentrated in whey protein. It does not exclude for example caseins. Further, to be consistent with our previous publication we strongly prefer to keep the term WPC.
5. Line 361-363 Moreover, the volume…of previous boluses: does this sentence refer to EH-WPC only or to both WPC’s?
We apologize that this information was not clear and have now indicated in line 376 that this sentence referred to EH-WPC.
6. Line 433: remove 1 m in ‘mmm’.
We have corrected this in the manuscript now.
7. Discussion: Is there any reason for concern around food safety of the milk with the reduced heating time of MP-WPC compared to EH-WPC or conventional milk processing? If so, it would be good to address this in the discussion.
There is no concern around food safety related to the MP-WPC as a pasteurization step was included in the process. From a food safety perspective, a pasteurization step is a requirement for infant formulas in order to prevent microbial/bacterial contamination. Specifically, the pasteurization of the WPC to obtain MP-WPC as used in the current study ensures the obtained product is sufficiently heat-treated with regards to prevention of microbial or bacterial contaminations and on the other hand ensures preservation of protein nativity.
8. I think the table S1 contains essential information that would be best to include in the manuscript – would the authors consider this?
Table S1 contains the protein composition of the MP-WPC and EH-WPC. We agree this is relevant information and have therefore put this now in the manuscript as Table 1.
Reviewer 3 Report
Line 95: what was the membrane cut-off?
Line 95: what was the WPC dissolved in?
Line 122: please add a ref or the company or describe the method for BCA.
line 224: remove "and"
line 249: is the pH during the gastric digestion usually around 4.0? If so please add a reference in the manuscript to indicate this. Is this the reason why SGF was used at pH 4.0?
Line 277: the sentence "whey proteins precipitate at pH4.". Is this a result from this study or does it refer to the study in ref 23? Is so it should be in the discussion. The same for ref 25. Table S1 refers to SDS-PAGE results so does this sentence relate to the SDS-PAGE analysis?
Line 277-278: The sentence "therefore...." is not clear. Please rephrase.
Figure 1 d and e: is this at pH 7.1 or 4.6? Please add it to the legend.
Line 320. Why were no feces on day 1,2 and 3? is this common? Please explain.
In general, there is a lack of discussion on near-term piglets results. It is not very clearly stated that there were no significant differences in most of the analysis.
Table S1: Please state the pH of the samples. How was the total protein calculated?
Author Response
We thank the author for taking the time to evaluate this manuscript and provide valuable feedback. We have implemented the comments as indicated in detail below:
Line 95: what was the membrane cut-off?
We apologize this relevant information was missing in the manuscript. Diluted skim milk was subject to cold microfiltration using a 0.08 µm pore size membrane. This is now added to the methods.
Line 95: what was the WPC dissolved in?
The WPC was dissolved in water before it was mildly pasteurized and freeze-dried again. We have added this information in line 100.
Line 122: please add a ref or the company or describe the method for BCA.
We have now added the appropriate reference.
line 224: remove "and"
We have removed this from the manuscript.
line 249: is the pH during the gastric digestion usually around 4.0? If so please add a reference in the manuscript to indicate this. Is this the reason why SGF was used at pH 4.0?
We aimed to mimic infant digestion conditions, which typically shows a pH around 4. A reference for this is now included in the methods section (ref [36], line 250).
Line 277: the sentence "whey proteins precipitate at pH4.". Is this a result from this study or does it refer to the study in ref 23? Is so it should be in the discussion. The same for ref 25. Table S1 refers to SDS-PAGE results so does this sentence relate to the SDS-PAGE analysis?
Line 277-278: The sentence "therefore...." is not clear. Please rephrase.
We realize the information in these sentences was a bit confusing and clarified this. The aim of this paragraph was to evaluate protein solubility and denaturation. We now state in line 283-291): “Extensive heating only mildly affected protein solubility at pH 7.1, with 98% soluble protein for MP-WPC compared with 87% for EH-WPC (Figure 1c), indicating less insoluble protein aggregates in the MP-WPC. Protein denaturation was highly affected by the extensive heat treatment, because at pH 4.6 72% soluble protein remained for MP-WPC compared with 21% for EH-WPC. As caseins and non-native whey proteins are known to precipitate at pH 4.6 [23], and as 24-28% of the protein in the WPC used in this study was β-casein (Table 1), our data reveal that the whey protein fraction in MP-WPC is close to 100% native and only 30% native in EH-WPC.”
Figure 1 d and e: is this at pH 7.1 or 4.6? Please add it to the legend.
We apologize this information was missing and have now added it to the legend. This data is at pH 7.1
Line 320. Why were no feces on day 1,2 and 3? is this common? Please explain.
It is often observed in this piglet model that there is little to no feces on day 1, 2 and 3 due to the minimal volume of enteral nutrition they receive. We have commented on this in line 333-334.
In general, there is a lack of discussion on near-term piglets results. It is not very clearly stated that there were no significant differences in most of the analysis.
We agree with your comment and now have implemented this limitation of the current study in the last section of the discussion (line 573-575).
Table S1: Please state the pH of the samples. How was the total protein calculated?
Upon request of another reviewer, Table S1 is moved to the main manuscript and is now Table 1. We have added that this was determined at pH 7.1, based on SDS-Page analysis. This is described in detail in methods section 2.2.